# Peri-Implant Behavior of Tissue Level Dental Implants with a Convergent Neck

**DOI:** 10.3390/ijerph18105232

**Published:** 2021-05-14

**Authors:** Rubén Agustín-Panadero, Irene Bermúdez-Mulet, Lucía Fernández-Estevan, María Fernanda Solá-Ruíz, Rocío Marco-Pitarch, Marina García-Selva, Álvaro Zubizarreta-Macho, Raquel León-Martínez

**Affiliations:** 1Department of Stomatology, Faculty of Medicine and Dentistry, University of Valencia, 46010 Valencia, Spain; ruben.agustin@uv.es (R.A.-P.); ibermu@alumni.uv.es (I.B.-M.); lucia.fernandez-estevan@uv.es (L.F.-E.); rocio.marco@uchceu.es (R.M.-P.); marina.garcia@uv.es (M.G.-S.); ralemar@alumni.uv.es (R.L.-M.); 2Department of Dentistry, Cardenal Herrera-CEU University, 46115 Valencia, Spain; 3Department of Implant Surgery, Faculty of Health Sciences, Alfonso X el Sabio University, 28691 Madrid, Spain; amacho@uax.es

**Keywords:** bone loss, bone level, tissue level, single crown, fixed partial prosthesis

## Abstract

Introduction: The aim of this retrospective study was to analyze the radiographic peri-implant bone loss of bone level implants and tissue level implants with a convergent neck in screw-retained single crowns and in screw-retained fixed partial prostheses, after two years of functional loading. Materials and methods: The sample was divided into two groups according to their type: Group I: supracrestal implants with convergent transmucosal neck; Group II: crestal implants. In each group we distinguish two subgroups according to the type of prosthetic restoration: single crowns and a three-piece fixed partial prosthesis on two implants. To quantify bone loss, parallelized periapical radiographs were analyzed at the time of implant placement and after two years of functional load. Results: A total of 120 implants were placed in 53 patients. After statistical analysis it was observed that for each type of implant bone loss was 0.97 ± 0.91 mm for bone level and 0.31 ± 0.48 mm for tissue level. No significant differences were found regarding the type of prosthesis and the location (maxilla or mandible) of the implants. Conclusions: Tissue level implants with a convergent transepithelial neck exhibit less peri-implant bone loss than bone level implants regardless of the type of prosthesis.

## 1. Introduction

In patients with one or more missing teeth, dental implant placement has been the treatment of choice for decades. The implant survival rate is about 95%, and this treatment modality is the most predictable and the safest option in such cases [1,2,3,4,5]. Osseointegration and marginal stability of the soft and hard tissues is important for assessing implant success. Traditionally, a bone loss of <1.5 mm around the implant during the first year has been considered to be physiological. However, this figure needs to be redefined, since recent studies have found that bone loss is approximately between 0.8 ± 0.4 mm [6,7], and is dependent upon a range of factors such as the morphology and geometry of the implant, the surface of the implant neck, the technique used to place the implant, the patient habits, and the type of prosthesis [5,7,8,9,10].

The implant can be placed at either a crestal level (bone level), with two interfaces (one between the implant and the transepithelial abutment, and another between the latter and the prosthesis, generating two microgaps) or at a supracrestal level (tissue level), with a single interface that acts as the junction between the implant and the prosthesis, and separates the microgap from the bone area reducing bacterial filtration in that zone [7,8,11,12,13]. Within supracrestal implants we find implants with a convergent transmucosal neck that present greater stability of the peri-implant tissues by leaving more space for the collagen fibers of the ligament to guarantee coronal migration and which were introduced following the Biological Oriented Preparation Technique (BOPT) philosophy proposed by Dr. Loi [14,15].

In addition to clinical exploration, a radiological evaluation is required in order to perform a correct follow-up of the implants, and therefore the bone level. Panoramic radiographs, cone-beam computed tomography (CBCT) and periapical radiographs with the paralleling technique are considered acceptable methods in this sense. It is advisable to perform controls at 3, 6 and 12 months, and then at least once a year [7,10,13,16,17,18].

The aim of this retrospective study was to analyze the radiographic peri-implant bone loss of bone level and tissue level implants with a convergent neck in screw-retained single crowns and in screw-retained fixed partial prostheses, after two years of functional loading.

Our working hypotheses were: H1—tissue level implants present less bone loss than bone level implants; H2—single crowns present less bone loss than three-unit bridges over two implants; and H3—there are no bone differences according to the implant position in the arch.

## 2. Material and Methods

### 2.1. Study Design

A retrospective study was conducted into the magnitude of peri-implant bone loss after two years of functional loading in tissue level implants with a convergent neck and bone level implants with a convergent neck (Figure 1), in two types of prosthetic restorations: single crowns and fixed partial prostheses (FPP). The patients were treated in the Dental Clinic of the University of Valencia (Valencia, Spain) between June 2016 and June 2017. The required data were collected after obtaining written informed consent from the participants. The study was approved by the local Clinical Research Ethics Committee (Ref. no. 1500285).

All the implants were placed by the same operator (R.A.-P.), who has been a surgeon for 15 years, in partially edentulous patients with missing teeth in the posterior zones (molars and premolars) of both the maxilla and the mandible. 

The study sample was divided into two groups of implants and four subgroups. Group I: PRAMA RF Sweden & Martina^®^ implants (Padua, Italy) presenting a machined neck with a height of 2.8 mm, straight morphology (0.8 mm) and hyperbolic profile (2 mm) with an internal hex connection (3.4 mm), leaving the neck at a supracrestal level (*n* = 60, tissue level) (Figure 2). Group II: PREMIUM/KHONO Sweden & Martina^®^ (Padua, Italy) with an internal hex connection (3.8 mm) and a peripheral ring at juxta-osseous level (*n* = 60, bone level) (Figure 3). In turn, two subgroups were established within each group. Group I_a_: tissue level implants restored with screw-retained single crowns (TSSC) (*n* = 30) (Figure 4); Group I_b_ tissue level implants restored with screw-retained fixed partial prostheses (TFPP) (*n* = 30) (Figure 5). Group II_a_: bone level implants restored with screwed single crowns (BSSC) (*n* = 30) (Figure 6); Group II_b_: bone level implants restored with screw-retained fixed partial prostheses (BFPP) (Figure 7) (*n* = 30).

### 2.2. Inclusion and Exclusion Criteria

The following inclusion criteria were applied: adult patients in good general health (ASA score I); patients carrying tissue level implants with a convergent neck (PRAMA RF, Sweden & Martina, Padua, Italy) or bone level implants with a convergent neck (KOHNO, Sweden & Martina, Padua, Italy) in posterior areas (molars and premolars) of the maxilla or mandible, and subjected to prosthetic restoration with single crowns or screw-retained three-unit bridges over two implants; non-smokers or smokers of <10 cigarettes a day; patients without parafunctions; patients with at least 24 months of follow-up after prosthetic loading and availability of the protocolized control periapical radiographs (at surgery, prosthetic loading, and control after one and two years).

The following exclusion criteria were applied: patients < 18 years of age; patients with poor oral hygiene, significant disease conditions or subjected to bisphosphonate treatment; smokers of >10 cigarettes a day; area of bone loss analysis presenting regeneration prior to implant placement; failure to report to the successive control visits during the first two years, and the absence of control radiographs.

### 2.3. Radiographic Technique and Bone Loss Measurement Method

The study information was retrieved from the database of the Dental Clinic of the University of Valencia (Valencia, Spain). The images were digital periapical radiographs obtained with a horizontal orientation, with the help of Rinn system positioners (Dentsply, York, PA, USA). The radiographs were digitalized, filed and processed with the Carestream^®^ RVG system (Atlanta, GA, USA). The measurements were made using the Rhinoceros^®^ application (Robert McNeel & Associates, Seattle, WA, USA), and all were obtained by a single observer (R.L.-M.) in order to avoid discrepancies in the radiographic interpretations.

Taking radiographic distortion into account, a known reference value was taken which in this case was the width of the prosthetic platform of the implant (in mm), as stated by the manufacturer. The program allows scaling of the image to the desired figure; accordingly, the image was adjusted to the width of the implant and the global real values referred to the implant, bone and bone loss.

After scaling of the image, height 0 was taken to represent the prosthetic platform of the implant in the case of group II (Figure 8) and the zone where the machined neck ends and the infrabony-treated part of the implant begins in group I (Figure 9). From this height, two perpendicular lines were traced: one mesial and the other distal, to the bone level or to the end of the radiolucent zone.

Analyses were conducted on the bone loss mesial and distal on the day of surgery and after two years of prosthetic loading.

### 2.4. Statistical Analysis

A descriptive analysis was conducted on the variable bone loss, together with an inferential analysis adopting a parametric approach. The Wald chi-square test was used, with a calculation of the corresponding 95% confidence interval (95% CI). The level of statistical significance was established at 5% (α = 0.05).

## 3. Results

A total of 120 implants were placed in 53 patients (mean age = 54.2 years, with 3 patients {20–30 years}; 21 patients {30–50 years} and 30 patients {50–75 years}): group I (tissue level, *n* = 60) and group II (bone level, *n* = 60). These two groups in turn were subdivided into group I_a_ (tissue level implants restored with single crowns (TSSC), *n* = 30) and group I_b_ (tissue level implants restored with fixed partial prosthesis (TFPP), *n* = 30); and group II_a_ (bone level implants restored with single crowns (BSSC), *n* = 30) and group II_b_ (bone level implants restored with fixed partial prosthesis (BFPP), *n* = 30). Forty-six implants were placed in the maxilla and 74 in the mandible (Figure 10).

The mean bone loss (MBL) after two years of functional loading was 0.64 ± 0.80 mm. In the analysis according to groups, we recorded an MBL of 0.97 ± 0.91 mm in group II and 0.31 ± 0.48 mm in group I, with a significantly lower MBL in the tissue level implants (*p* < 0.001; Wald chi-square test).

In the analysis of bone loss according to the type of prosthesis, MBL in the crowns was 0.55 ± 0.68 mm versus 0.73 ± 0.90 mm in the FPP. No statistically significant differences were observed (*p* = 0.338; Wald chi-square test). When analyzing the influence of the type of prosthesis and implant upon bone loss, MBL was 0.24 ± 0.40 mm in group I_a_ (TSSC) and 0.37 ± 0.55 mm in group I_b_ (TFPP) and the difference was nonsignificant (*p* = 0.490; Wald chi-square test). In turn, MBL was 0.86 ± 0.7 mm in group II_a_ (BSSC) and 1.09 ± 1.04 mm in group II_b_ (BPPF) and the difference was also nonsignificant (*p* = 0.274; Wald chi-square test).

In the analysis of the peri-implant bone loss according to the type of implant, the bone level implants were found to be associated with greater bone loss than the tissue level implants (*p* < 0.001; Wald chi-square test); this difference was seen to be the same in the group of crowns and FPP (*p* < 0.001; generalized estimating equations (GEE)).

In the implants subjected to restoration with FPP, we found MBL to be significantly greater in the anterior zone versus the posterior zone: 0.92 ± 1.14 mm and 0.54 ± 0.52 mm, respectively (*p* < 0.010; Wald chi-square test). When comparing this result in both groups, we found group II_b_ (BPPF) to show significantly greater bone loss in the anterior versus the posterior implants, with MBL being 1.44 ± 1.32 mm and 0.74 ± 0.47 mm, respectively (*p* = 0.001; Wald chi-square test). However, in group I_b_ (TFPP) the bone loss in anterior versus posterior implants was 0.40 ± 0.62 mm and 0.35 ± 0.50 mm and the difference was nonsignificant (*p* = 0.606; Wald chi-square test).

In relation to bone loss according to the implant location in the maxilla and mandible, the MBL was found to be 0.46 ± 0.63 mm and 0.75 ± 0.88 mm, respectively. Despite the tendency towards an increased bone loss in mandibular implants, the differences were not statistically significant (*p* = 0.079; Wald chi-square test). Considered separately, group I exhibited a mean loss of 0.25 ± 0.50 mm in the maxilla versus 0.35 ± 0.47 mm in the mandible and the difference was nonsignificant (*p* = 0.184; test Chi^2^ of Wild), while group II showed a mean loss of 0.76 ± 0.68 mm in the maxilla versus 1.07 ± 1 mm in the mandible. This difference was also not statistically significant (*p* = 0.476; Wald chi-square test).

When assessing the interaction between the type of implant and the position in the dental arch of the single crowns, only the type of implant (bone level or tissue level) was seen to present significant differences in terms of peri-implant bone loss (*p* < 0.001; Wald chi-square test).

In the case of implants restored with fixed partial prostheses, bone loss was analyzed by evaluating the influence of the different variables jointly: type of implant, location and position corresponding to the fixed partial prosthesis. The type of implant proved significant (*p* < 0.001) since the bone level implants developed greater bone loss than the tissue level implants. A significant effect was also observed for the position (anterior or posterior) (*p* = 0.001) since the anterior implants showed greater bone loss than the posterior implants. This greater loss associated with anterior implants was only detectable in the bone level implants, however (*p* = 0.002; Bonferroni correction) (Figure 11).

The triple interaction among type of implant, type of prosthesis and position of the implant in the dental arch was statistically significant (*p* < 0.001), with the tissue level implants exhibiting lower bone loss in all cases (Figure 12).

## 4. Discussion

The present study was carried out in 53 patients carrying 60 bone level implants and 60 tissue level implants. The literature contains articles with similar samples and objectives, such as the studies by López et al. and Wallner et al. [11,12,19,20]. Radiography was the most common tool for assessing peri-implant bone loss, with acceptable options being CBCT, panoramic radiographs and periapical radiographs with the paralleling technique [16,17]. In our study, we made use of periapical radiographs involving the paralleling technique, with the help of Rinn system positioners (Dentsply, York, PA, USA). Other authors have used the same method as this approach is more precise and does not have the inconveniences of superposition, deformation or magnification of the images seen with the use of extraoral radiographs [11,19,20].

The Rhinoceros^®^ application (Robert McNeel & Associates, Seattle, USA) was used to measure bone loss, allowing us to scale the images and measure mesial and distal radiographic bone loss. Other authors have adopted the same method [8,13,21,22].

Following the statistical analysis of bone loss measured from the periapical radiographs, it can be affirmed that the tissue level implants generated significantly less peri-implant bone loss than the bone level implants (*p* < 0.001; Wald chi-square test). Consensus is lacking in the literature regarding the association between the type of implant and bone loss. Agustin-Panadero et al. analyzed six groups of implants (three tissue level and three bone level implant groups) and recorded significantly lower bone loss in all the tissue level implant groups [13]. Bilhan et al. compared bone loss with 105 tissue level implants versus 36 bone level implants, and likewise found lower losses in the former group [23]. However, other studies have recorded no statistically significant differences [11,12,24,25]. This difference could be due to the location of the microgap between the prosthetic connection and the transepithelial abutment with respect to the bone crest. In the case of the tissue level implants, this zone is located away from the bone, preventing bacterial penetration of the bone crest. In contrast, in the bone level implants, the microgap is located at bone level, which may result in bacterial filtration with subsequent inflammation and greater bone loss [26]. The morphology of the implant neck has also been shown to be very important for the stability of the peri-implant tissues. Two recent meta-analyses have demonstrated that implants with convergent necks present less bone loss compared to implants with divergent neck morphology, which is due, among other factors, to the fact that there is a greater space for the accommodation of soft tissues, showing that a soft tissue > 2 mm guarantees less bone resorption [14,27]. This hypothesis was originally analyzed in histological studies in dogs where they concluded that the geometry of the abutment influences the orientation of the circular collagen fibers and this, in turn, on peri-implant bone stability [28].

Regarding the difference in bone loss in single implants restored with crowns or multiple implants restored with FPP, the results revealed no statistically significant differences. These findings are consistent with those of other studies such as that published by Firme et al., who conducted a meta-analysis comparing bone loss between these two types of prostheses and recorded no statistically significant differences, as well as with the data from other recent studies [29]. Nevertheless, crowns presented a certain tendency towards lower bone loss that may be due to more favorable emergence profiles, better passive insertion of the prosthesis, and better oral hygiene [21,30].

In relation to the implants in our study restored with FPP, we evaluated possible differences in bone loss in anterior implants versus implants located in posterior zones. Both overall and within the crestal implants, bone loss was seen to be greater in the anterior implants than in the implants located in posterior zones. This observation was consistent with the data reported by Sola-Ruiz et al., who found that anterior implants resulted in greater bone loss [21].

Regarding the differences in MBL according to the dental arch involved, consensus is lacking in the literature, since some studies describe greater bone loss in the maxilla due to a lack of cortical bone stabilization and overall bone volume, and lower trabecular bone density, with respect to the mandible [31]. In contrast, other investigators have observed no statistically significant differences [32,33].

Thus, regarding our working hypotheses, we can accept H1, since the tissue level implants presented lower bone loss than the bone level implants. H2 is rejected, since there were no differences in MBL between crowns or FPP. Lastly, H3 is accepted, since the results revealed no bone differences in relation to the position of the implant in the arch.

## 5. Conclusions

Tissue level implants with a convergent transepithelial neck exhibit less peri-implant bone loss than bone level implants. The type of prosthesis over the implants does not influence peri-implant bone loss, though implants located in an anterior position within a fixed partial prosthesis are associated with greater bone loss than those located in a posterior position in the mouth. Implants placed in the maxilla or mandible present similar peri-implant bone loss values.

## Figures and Tables

**Figure 1 ijerph-18-05232-f001:**
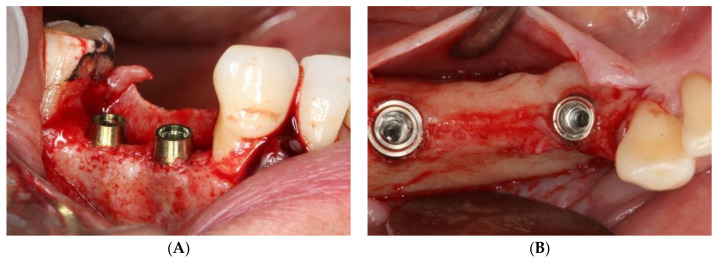
(**A**) tissue level implants, (**B**) bone level implants.

**Figure 2 ijerph-18-05232-f002:**
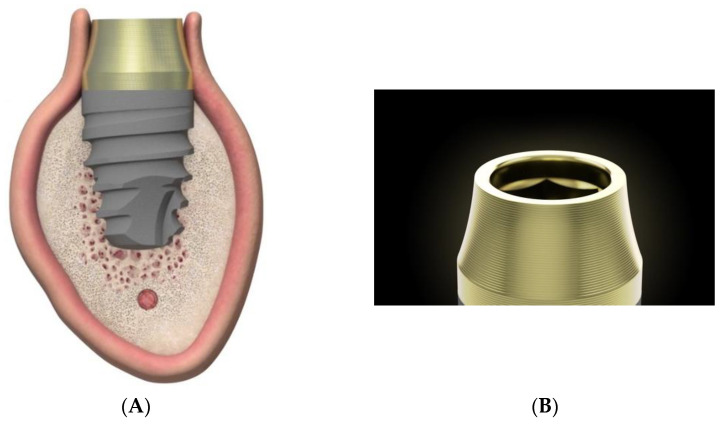
(**A**) lateral image of the design of the PRAMA implant in relation to the peri-implant tissues. (**B**) morphology of the neck of the PAMA implant.

**Figure 3 ijerph-18-05232-f003:**
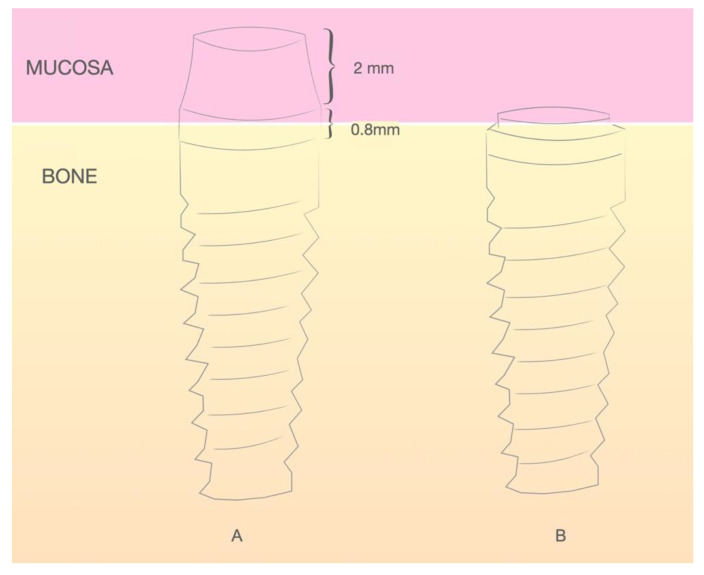
(**A**) ubication of the PRAMA implant with respect to the peri-implant tissue. (**B**) ubication of the KHONO implant with respect to the peri-implant tissue.

**Figure 4 ijerph-18-05232-f004:**
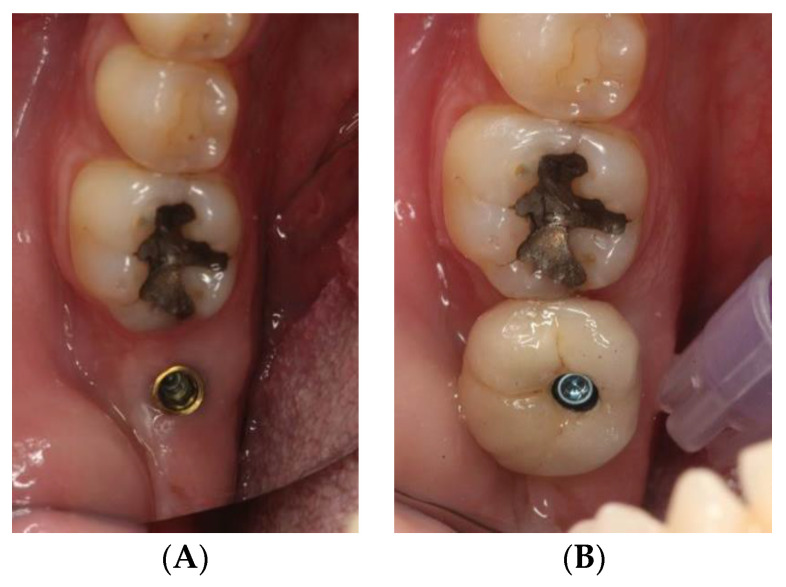
(**A**) implant tissue level PRAMA RF in position 3.7, (**B**) implant tissue level PRAMA RF restored with a screw-retained single crown.

**Figure 5 ijerph-18-05232-f005:**
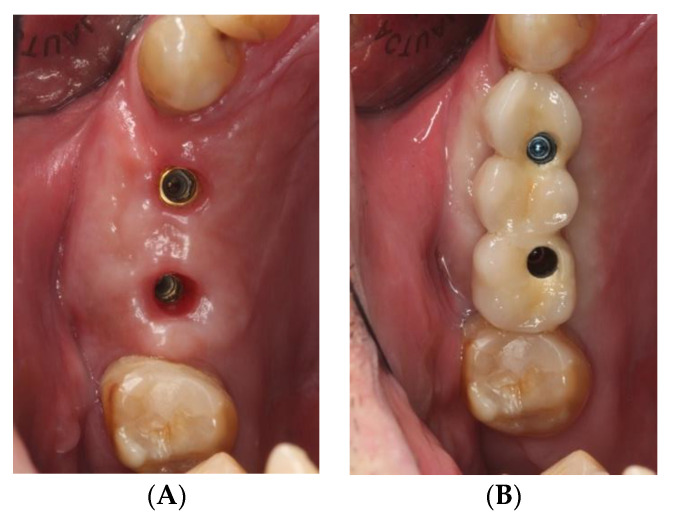
(**A**) two tissue level implants PRAMA RF in position 1.4 and 1.6, (**B**) fixed partial prosthesis (3 pieces) on 2 tissue level implants PRAMA RF.

**Figure 6 ijerph-18-05232-f006:**
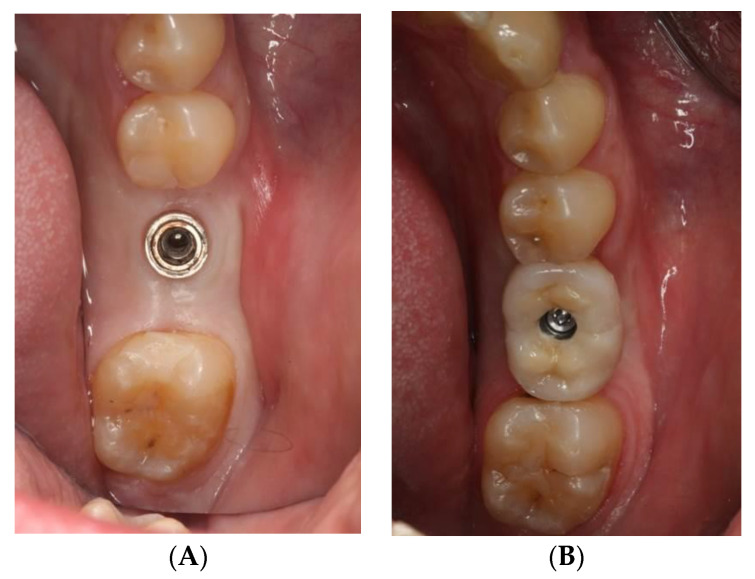
(**A**) implant bone level KOHNO in position 4.6, (**B**) implant tissue level KOHNO restored with a screw-retained single crown.

**Figure 7 ijerph-18-05232-f007:**
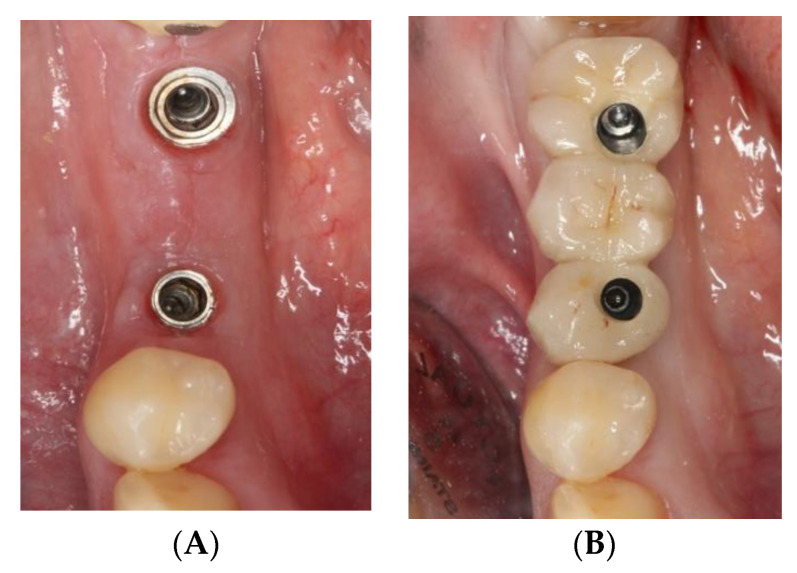
(**A**) two bone level implants KOHNO in position 3.5 and 3.7, (**B**) fixed partial prosthesis (3 pieces) on 2 bone level implants KOHNO.

**Figure 8 ijerph-18-05232-f008:**
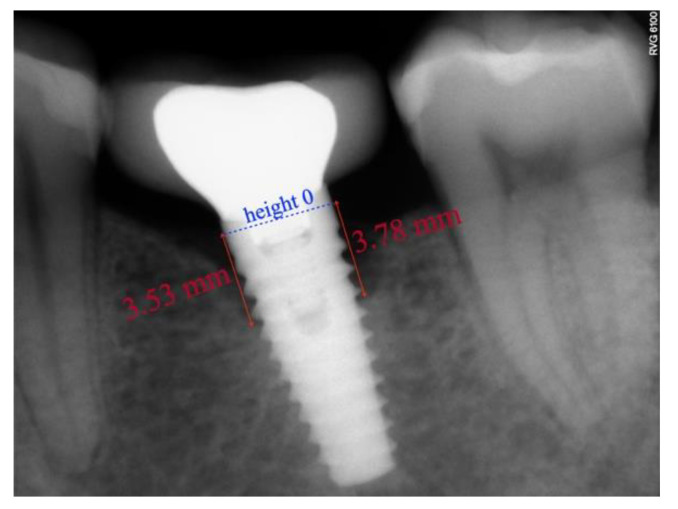
Periapical radiograph of a crestal implant (KOHNO) after 24 months of prosthetic loading. Values in red indicate the bone loss that has occurred.

**Figure 9 ijerph-18-05232-f009:**
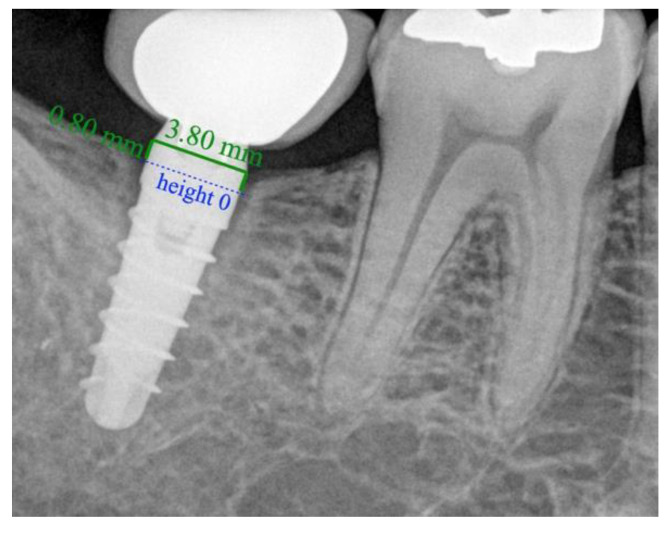
Periapical radiograph after 24 months of prosthetic loading of a supracrestal implant with convergent neck (PRAMA RF) without bone loss.

**Figure 10 ijerph-18-05232-f010:**
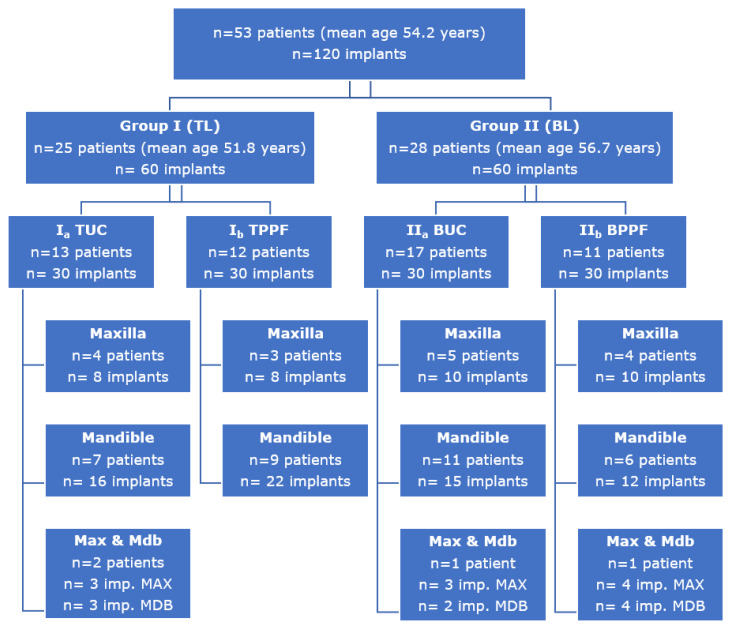
Distribution of the sample according to implant level and the type and location of the restoration.

**Figure 11 ijerph-18-05232-f011:**
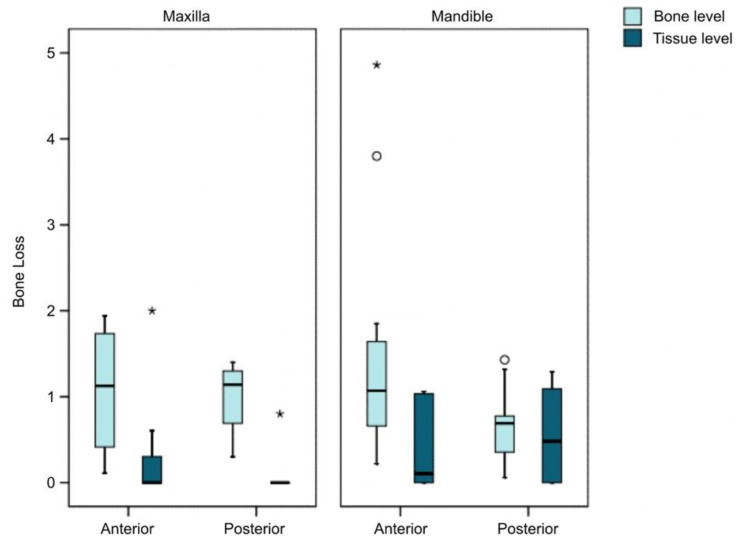
Bone loss according to implant type and position in the arch in FPP. * major outlier ° minor outlier.

**Figure 12 ijerph-18-05232-f012:**
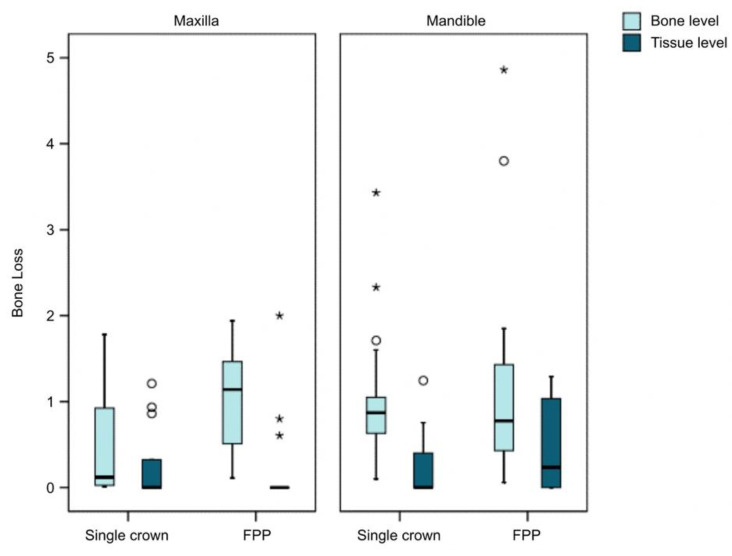
Bone loss according to type of prosthesis and implant. * major outlier ° minor outlier.

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
