# Peer review of "Peri-Implant Behavior of Tissue Level Dental Implants with a Convergent Neck"

_ijerph, 2021, doi:10.3390/ijerph18105232_

Round 1
Reviewer 1 Report
Dear authors,
I very much enjoyed reading this study. I am very familiar with this type of tissue level implants object of this study and I believe that it is essential to study both their functional mechanical and biological aspects.
I don't have much to comment on the methodological aspects of the research which has all the limitations of a retrospective study but nevertheless deserves to be published.
However, I would underline the concept behind a convergent abutment, namely that of leaving more space for the soft tissues and ensuring an upward, rather than downward, migration of the collagen fibers of the peri-implant circular ligament. In this regard, it is essential to mention a recent systematic review that deals precisely with the comparison between convergent versus divergent (or parallel) abutments: PMID: 32956437
At the base of this concept of convergent abutment that led to the creation of the implants studied in the article is the intuition of Dr. Loi who created the biologically oriented preparation technique (BOPT), which applied to the implants led to the creation of the PRAMA, I believe it is essential to give credit by citing at least the most representative article of this technique: PMID: 23390618.
Leaving more space for soft tissues means preserving hard tissues, in fact, as demonstrated, soft tissues thicker than 2mm guarantee less bone resorption. In this regard, I would comment in the discussions and cite the latest systematic review published in this regard: PMID: 32643328
Finally, in the discussions, you might want to comment on an interesting animal study that compares convergent profile implants with divergent profile implants: PMID: 24118965
Congratulations on your work
Best regards
Reviewer 2 Report
Review on the article Peri-implant Behavior of Tissue Level Dental Implants with a Convergent Neck, submitted to International Journal of Environmental Research and Public Health.
The article is well-written in terms of presenting current data of the reference, the aim of the research, materials and methods and results with discussion, as well as following conclusions. However, there are some minor comments that can increase the objectivity of the data presented as well as the quality of the data presented. All comments in accordance to each section are presented below.
- Abstract
There is no critical comment about this section.
- Introduction
The introduction is properly written with the use of appropriate number of the references that present current state-of-the-art.
- Materials and Methods
Suggestions:
- Combine Figure 1 and 2 into a single Figure divided into part A and part B. Do the same with pairs of Figure 3 and 4, 5 and 6, 7 and 8, 9 and 10.
- Provide short information about the experience of the surgeon performing the implantation process – this can confirm that the operations were appropriately performed.
- Provide an additional Figure presenting a scheme of the implant used (include overall dimensions of the construction) and its placement into the bone tissue. This is especially important as tissue reaction depends on the shape of the implant.
The authors are encouraged to provide age of the patients included in the research (min. max. and mean). Age is one of the factors that has an influence on the bone tissue quality. I would suggest presenting, despite of mean age, the number of participants in intervals (i.e. 18-30 years: 3 patients, 30-40 years: 5 patients, etc.). This would give appropriate insight into the age of participants.
- Results
The scheme presented under Table 1 heading is a Figure not a table. This should be clarified. Moreover, this graph should include the data of patients’ mean age etc.
- Discussion
There is no critical comment about this section.
- Conclusions
There is no critical comment about this section.
Author Response
Please see the attchment
